# MISSDEEPCAUSAL: CAUSAL INFERENCE FROM INCOMPLETE DATA USING DEEP LATENT VARIABLE MODELS

## ABSTRACT

Inferring causal effects of a treatment, intervention or policy from observational data is central to many applications. However, state-of-the-art methods for causal inference seldom consider the possibility that covariates have missing values, which is ubiquitous in many real-world analyses. Missing data greatly complicate causal inference procedures as they require an adapted unconfoundedness hypothesis which can be difficult to justify in practice. We circumvent this issue by considering latent confounders whose distribution is learned through variational autoencoders adapted to missing values. They can be used either as a pre-processing step prior to causal inference but we also suggest to embed them in a multiple imputation strategy to take into account the variability due to missing values. Numerical experiments demonstrate the effectiveness of the proposed methodology especially for non-linear models compared to competitors.
Keywords: treatment effect estimation, missing values, variational autoencoders, importance sampling, double robustness, multiple imputation.

## 1 INTRODUCTION

Many methods have been developed to estimate the causal effect of an intervention, such as the administration of a treatment, on an outcome such as survival, from observational data, i.e., data that is potentially confounded by selection bias due to the absence of randomization. Classical ones include matching (Iacus et al., 2012), inverse propensity weighting (IPW, Horvitz & Thompson, 1952; Rosenbaum & Rubin, 1983) and doubly robust methods (Robins et al., 1994; Chernozhukov et al., 2018; Wager & Athey, 2018; Athey et al., 2019). More recent proposals use deep learning methods that ensure balance of the population at the level of representation (Johansson et al., 2016; Shalit et al., 2017), infer the joint distribution of latent and observed confounders, the treatment and the outcome (Louizos et al., 2017) or predict the counterfactuals with GANs (Yoon et al., 2018). For a detailed review of existing literature on treatment effect estimation we refer to Imbens (2004), Lunceford & Davidian (2004) and Guo et al. (2019).

However, state-of-the-art methods still suffer from important shortcomings. In particular, they seldom consider the possibility that covariates have missing values, which is ubiquitous in many real-world situations (Josse & Reiter, 2018) and has been widely discussed in different contexts (Mayer et al., 2019a; van Buuren, 2018; Little & Rubin, 2002). Although this question of missing attributes in the context of treatment effect estimation has been raised early in the development of causal inference (Rosenbaum & Rubin, 1984), there is still a lack of effective and consistent solutions addressing this problem, with a few notable exceptions such as Mattei & Mealli (2009); Seaman & White (2014); Yang et al. (2019); Kallus et al. (2018) which mainly focus on inverse propensity weighting (IPW) methods and Kuroki & Pearl (2014) who discuss identifiability of causal effects under measurement error or unobserved confounders. Recently, Mayer et al. (2019b), in addition to suggesting doubly robust estimators with missing data, classified the existing approaches into two families: the ones that adapt the causal inference assumptions to the missing values setting (D'Agostino Jr & Rubin, 2000; Blake et al., 2019) and the ones (Mattei & Mealli, 2009; Seaman & White, 2014; Kallus et al., 2018) that consider the classical machinery and missingness mechanisms assumptions (Little & Rubin, 2002). While the former are based on the assumption of *unconfoundedness with missing values*, which can be difficult to assess in practice, the latter have been developed under strong para-

metric assumptions about the outcome, treatment and covariates models, in addition to relying on missing values hypotheses that can also be difficult to meet in practice (Yang et al., 2019).

To avoid relying on the hypothesis of unconfoundedness with missing values or being in the very parametric (and linear) framework of multiple imputation (Mattei & Mealli, 2009; Seaman & White, 2014) and matrix factorization (Kallus et al., 2018), we propose a new method for causal inference with missing data, which we call MissDeepCausal. MissDeepCausal is inspired by the work of Kallus et al. (2018) in the sense that we consider a model with latent confounders, and assume that we only have access to covariates with missing values that are noisy proxies of the true latent confounders. However, our approach generalizes and extends the work of Kallus et al. (2018) in different aspects: (i) instead of linear factor analysis models with missing values, we consider non-linear versions using deep latent variable models (Kingma & Welling, 2014; Rezende et al., 2014); (ii) we rely on the missing at random (MAR) (Rubin, 1976) assumption for the missing data mechanisms, and not on the stronger missing completely at random (MCAR) one; (iii) we take into account the posterior distribution of the latent variables given observed data and not only their conditional expectation. This latter point allows us to define a multiple imputation strategy adapted to the latent confounders model, and to couple it with doubly robust treatment effect estimation (Chernozhukov et al., 2018).

In the remainder of this article we first introduce the problem framework and recall existing work for handling missing values in causal inference in Section 2. We then introduce two variants of our MissDeepCausal approach in Section 3. Finally we compare MissDeepCausal empirically with several state-of-the-art methods on simulated data in Section 4.

## 2 SETTING, NOTATIONS AND RELATED WORKS

In this section we start by quickly reviewing the problem of causal inference from observational data without missing data. We consider the potential outcomes framework (Rubin, 1974; Imbens & Rubin, 2015) where we have a sample of $n$ independent and identically distributed (i.i.d.) observations $(Y_i(0), Y_i(1), W_i, X_i)_{i=1, \dots, n}$ with $W_i \in \{0, 1\}$ a binary treatment, $X_i = (X_{i1}, \dots, X_{ip})^\top \in \mathbb{R}^p$ a vector of covariates, and $(Y_i(0), Y_i(1)) \in \mathbb{R}^2$ the outcomes we would have observed had we assigned control or treatment to the $i$-th sample, respectively. The observed outcome for unit $i$, $Y_i \in \mathbb{R}$ is defined as $Y_i \triangleq W_i Y_i(1) + (1 - W_i)Y_i(0)$. The individual causal effect of the treatment is $\tau_i \triangleq Y_i(1) - Y_i(0)$ and the average treatment effect (ATE) is defined as

$$\tau \triangleq \mathbb{E}[Y_i(1) - Y_i(0)] = \mathbb{E}[\tau_i].$$

The ATE $\tau$, i.e., the link between $W$ and $Y$, can be estimated from observational data by taking into account the confounding factors $X$, i.e., the common causes of $W$ and $Y$. A popular estimator of $\tau$ from observational data is the so-called doubly robust estimator:

$$\hat{\tau}_{DR} \triangleq \frac{1}{n} \sum_{i=1}^{n} \hat{\mu}_1(X_i) - \hat{\mu}_0(X_i) + W_i \frac{Y_i - \hat{\mu}_1(X_i)}{\hat{e}(X_i)} - (1 - W_i) \frac{Y_i - \hat{\mu}_0(X_i)}{1 - \hat{e}(X_i)}, \qquad (1)$$

where $\hat{\mu}_w(x)$ are regression estimates of the conditional response surfaces $\mu_w(x) \triangleq \mathbb{E}[Y(w) \,|\, X = x]$, $w \in \{0, 1\}$, and $\hat{e}(x)$ is an estimate of the *propensity score* $e(x) \triangleq \mathbb{P}(W_i = 1 \,|\, X_i = x)$ (Rosenbaum & Rubin, 1983; Imbens & Rubin, 2015).

Standard results state that if either $(\hat{\mu}_0, \hat{\mu}_1)$ or $\hat{e}$ is correctly specified, then $\hat{\tau}_{DR}$ is an unbiased estimator of $\tau$ (Robins et al., 1994; Chernozhukov et al., 2018; Wager & Athey, 2018) under the following assumptions (Rosenbaum & Rubin, 1983): the *ignorability* or *unconfoundedness* assumption that states that all confounding factors are measured, i.e., conditionally on $X$, the treatment assignment is independent of the potential outcomes:

$$\{Y_i(1), Y_i(0)\} \perp\!\!\!\perp W_i \,|\, X_i, \qquad \text{for all } i; \qquad (2)$$

and the *overlap* assumption assuming the existence of some $\eta > 0$ such that $\eta < e(x) < 1 - \eta$, for all $x \in \mathcal{X}$.

We now consider an extension to account for possible missing entries in the covariates. For that purpose, we denote the missingness pattern of the $i$-th sample as $M_i \in \{0, 1\}^p$ such that $M_{ij} = 0$

if $X_{ij}$ is observed and $M_{ij} = 1$ otherwise. The matrix of observed covariates can be written as $X^\star \triangleq X \odot (\mathbf{1} - M) + \text{NA} \odot M$, with $\odot$ the elementwise multiplication and $\mathbf{1}$ the matrix filled with 1, so that $X^\star$ takes its value in the half discrete space $\mathcal{X}^\star \triangleq (\mathbb{R} \cup \{\text{NA}\})^p$. We model $M_i$ as a random vector, and the possibility to infer causal effects with missing data now depends on additional assumptions on the joint law of $(Y_i(0), Y_i(1), W_i, X_i, M_i)_{i=1,\dots,n}$. Methods for causal inference with missing covariates can be classified into two categories.

**Unconfoundedness with missing values.** Rosenbaum & Rubin (1984) extend the unconfoundedness hypothesis (2) to missing values as

$$\{Y_i(1), Y_i(0)\} \perp\!\!\!\perp W_i \,|\, X_i^\star, \qquad \text{for all } i. \tag{3}$$

This implies the assumption, illustrated in Figure 1, that if a covariate is not observed, it is not a confounder. In particular, observations can have different confounders depending on their pattern of missing data. They define the generalized propensity score as:

$$\forall x^\star \in \mathcal{X}^\star, \quad e^\star(x^\star) \triangleq \mathbb{P}(W_i = 1 \,|\, X_i^\star = x^\star), \tag{4}$$

which is a balancing score under (3). Consequently, an IPW estimator formed with estimators of $e^\star$ can be an unbiased estimator of the ATE with missing values. Nevertheless, this method relies both on the fact that the covariates $X$ are the appropriate set of confounders, which can be questioned without missing data (Kallus et al., 2018), and requires certain expert input and reasoning to verify that for each observation, treatment assignment and/or outcome values depend only on observed values of the confounders (Blake et al., 2019; Mayer et al., 2019b). Note in particular, that it is not because the missing data in the covariates are completely at random (MCAR), i.e., $M \perp\!\!\!\perp X$, that (3) is met. In practice, in addition, a difficulty with this approach is that estimating (4) requires fitting one model per pattern of missing values, which is unrealistic with classical tools (Miettinen, 1985; D'Agostino Jr & Rubin, 2000; D'Agostino Jr et al., 2001; Blake et al., 2019); Mayer et al. (2019b) address this problem using random forests adapted to covariates with missing values.

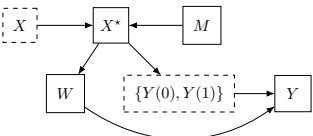

Figure 1: Unconfoundedness with missing values. $X$ represents a the complete covariates, and $M$ a missing data mechanism, $X^*$ represents the observed incomplete covariates, confounding the treatment assignment. The formalism of Pearl (1995) and Richardson & Robins (2013) is used.

**Missingness mechanisms assumptions.** Multiple imputation is one of the most powerful approaches to estimate parameters and their variance from an incomplete data (Little & Rubin, 2002; van Buuren, 2018). Seaman & White (2014) show that when assuming (i) identifiability of the ATE in the complete case, (ii) missing at random (MAR) values given $W$ and $Y$, (iii) correct specification of the propensity score with logistic regression and of the Gaussian distribution of covariates, then multiple imputation gives a consistent estimate for the ATE estimated with IPW. An extension to doubly robust estimation has been proposed by Mayer et al. (2019b).

Instead of assuming that confounders are observed directly, Kallus et al. (2018) consider a more general model where observed covariates $X$ are noisy and/or incomplete proxies of the true latent confounders $Z$. More specifically, they assume a low-rank model for the covariates and estimate the latent variables from the incomplete confounders using matrix completion methods (Hastie et al., 2015; Josse et al., 2016). Then, under the linear regression model

$$Y_i = Z_i^T \alpha + \tau W_i + \varepsilon_i, \tag{5}$$

with random latent variables $Z$, missing values completely at random (MCAR) in $X$, unconfoundedness given $Z$, and some additional assumptions, they prove that regressing $Y$ on $\hat{Z}$ and $W$ leads to a consistent ATE estimator. Both techniques, multiple imputation and matrix factorization, rely on parametric (and linear) frameworks.

## 3 MISSDEEPCAUSAL

To avoid relying on the hypothesis of unconfoundedness with missing values (3) or being in the very parametric (and linear) framework of multiple imputation and matrix factorization, we propose MissDeepCausal, an approach based on deep latent variable models where the latent variables are assumed to be the confounders as represented in Figure 2.

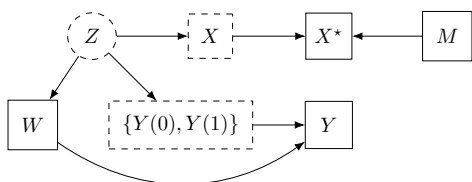

Figure 2: Graphical representation of the model underlying MissDeepCausal. $Z$ represents the unobserved latent confounders of the treatment $W$ and the effect $Y$. $X$ represents a proxy for the confounders, and $M$ a missing data mechanism; $X^*$ represents the observed incomplete covariates.

Under this model, the unconfoundedness hypothesis (3) does *not* hold, so a standard treatment effect estimator using $X^*$ as covariates would be biased. On the other hand, we can express the treatment effect conditioned on $X^*$ as follows:

$$\mathbb{E}[Y(1) - Y(0) \,|\, X^*] = \mathbb{E}[\mathbb{E}[Y(1) - Y(0) \,|\, Z, X^*] \,|\, X^*]$$
$$= \mathbb{E}[\mathbb{E}[Y(1) - Y(0) \,|\, Z] \,|\, X^*].$$

Consequently, if we have an unbiased estimator $\hat{f}(Z)$ of $\mathbb{E}[Y(1) - Y(0) \,|\, Z]$, the treatment effect conditioned on $Z$, and if we know $P(Z \,|\, X^*)$, the conditional distribution of $Z$ given $X^*$, then we can derive the treatment effect conditioned on $X^*$ by

$$\hat{g}(X^\star) \triangleq \mathbb{E}[\hat{f}(Z)|X^\star]. \tag{6}$$

Furthermore, by expressing the ATE as

$$\tau = \mathbb{E}[Y(1) - Y(0)] = \mathbb{E}[\mathbb{E}[Y(1) - Y(0) \,|\, X^*]],$$

we can form an estimate of the ATE by $\mathbb{E}[\hat{g}(X^\star)]$. We describe such an estimator in Section 3.2 below, which is reminiscent of multiple imputation techniques in the field of missing value imputation (Rubin, 1987).

Another strategy, described in Section 3.3, is to consider latent variables estimation as a pre-processing step prior to causal inference by computing

$$h(X^\star) \triangleq f(\mathbb{E}[Z|X^\star]); \tag{7}$$

this can be seen as a non-linear extension of Kallus et al. (2018). Both estimators require sampling from the posterior distribution $P(Z|X^\star)$. Consequently, we first describe in Section 3.1 how to learn the joint distribution of $(Z, X)$ from $X^\star$ using a variational autoencoder (VAE) with missing data, before turning to the details of each strategy.

### 3.1 DEEP LATENT VARIABLE MODELS WITH MISSING VALUES

To estimate and sample from $P(Z \,|\, X^\star)$, we use the missing data importance weight autoencoder bound (MIWAE) approach of Mattei & Frellsen (2019), which is summarized in the appendix A. They use a simple variational family where they impute the missing entries with a constant and show that using this class of distributions, it maximizes a lower bound of the observed log-likelihood. Note that their approach requires the classical missing at random (MAR) (Rubin, 1976) assumption to ignore the missing values mechanism when maximizing the observed likelihood for the VAE inference.

In the MIWAE approach, the variational distribution $Q_\gamma(Z|X^\star)$ (defined in Appendix A) plays a central role but is not necessarily a good surrogate for the posterior distribution $P_\theta(Z|X^\star)$. To

sample from the true posterior distribution, we resort to importance sampling techniques using the variational distribution $Q_\gamma$ for proposal. More precisely, we can define, for any measurable function $s$,

$$\mathbb{E}[s(Z)|X^\star] = \int s(Z)p_\theta(Z|X^\star)dZ = \frac{1}{p(X^\star)} \int s(Z)\frac{p_\theta(X^\star|Z)p(Z)}{q_\gamma(Z|X^\star)}q_\gamma(Z|X^\star)dZ.$$

This quantity can be estimated using self-normalized importance sampling with:

$$\mathbb{E}[s(Z)|X^\star] \approx \sum_{l=1}^{L} w_l s(Z^{(l)}), \text{ where } w_l \triangleq \frac{r_l}{r_1 + ... + r_L}, \text{ with } r_l \triangleq \frac{p_\theta(X^\star|Z^{(l)})p(Z^{(l)})}{q_\gamma(Z^{(l)}|X^\star)}. \quad (8)$$

Equation (8) is used in our second strategy described in Section 3.3, while for our first strategy (described in Section 3.2) we sample $L$ samples $Z^{(1)}, \ldots, Z^{(L)}$ according to $Q_\gamma(Z|X^\star)$, compute the weights as in (8) and re-sample $B << L$ with probability proportional to the weights.

### 3.2 MISSDEEPCAUSAL WITH MULTIPLE IMPUTATION (MDC-MI)

MDC-MI uses the importance sampling strategy presented in Section 3.1, to compute an approximation of (6) by Monte-Carlo as follows. First, we draw $B$ i.i.d. samples $(Z^{(j)})_{1 \leq j \leq B} \in \mathbb{R}^{n \times d}$ from the posterior distribution $P(Z|X^\star)$. On each sample, we evaluate the function $\tilde{f}$ and aggregate the results: $\hat{g}^{(B)}(X^\star) = \frac{1}{B} \sum_{j=1}^{B} f(Z^{(j)})$. This approach can be viewed as a multiple imputation method, which consists in generating different imputed data sets by drawing the missing values from their posterior distribution given observed values, then estimating the parameters of interest on each imputed data set and aggregating the results according to Rubin's rules (Rubin, 1987) to obtain a final estimate for the quantity of interest. Here we consider the samples $Z^{(j)}$ of the latent variables and apply the doubly robust estimator from (1) on each table $Z^{(j)}$:

$$\hat{\tau}^{(j)} = \frac{1}{n} \sum_{i=1}^{n} \hat{\mu}_1^{(j)}(Z_i^{(j)}) - \hat{\mu}_0^{(j)}(Z_i^{(j)}) + W_i \frac{Y_i - \hat{\mu}_1^{(j)}(Z_i^{(j)})}{\hat{e}^{(j)}(Z_i^{(j)})} - (1 - W_i)\frac{Y_i - \hat{\mu}_0^{(j)}(Z_i^{(j)})}{1 - \hat{e}^{(j)}(Z_i^{(j)})}, \quad (9)$$

and get the final estimate for the causal effect by computing the mean of the estimators i.e. $\hat{\tau} = \frac{1}{B} \sum_{j=1}^{B} \hat{\tau}^{(j)}$. The doubly robust estimator from (1) is asymptotically normal (under some mild assumptions) (Wager & Athey, 2018) which is required for the aggregation in multiple imputation procedures (Rubin, 1987). Note that this multiple imputation strategy additionally allows to reflect the variability due to the missing values in the variance estimation of the estimator $\hat{\tau}$.

### 3.3 MISSDEEPCAUSAL WITH LATENT VARIABLES ESTIMATION AS A PRE-PROCESSING STEP (MDC-PROCESS)

We also propose MDC-process as a non-linear extension of Kallus et al. (2018), where we estimate $h(X^\star)$ defined in (7). For that purpose, we first approximate the expectation of the posterior distribution

$$\hat{Z}(x^\star) \triangleq \mathbb{E}[Z|X^\star = x^\star] \quad (10)$$

to get estimates for the latent confounders. In a second step, we use them under the regression model (5) and accordingly regress the observed outcome $Y$ on the estimated latent factors $\hat{Z}(x^\star)$ and the treatment assignment $W$ to obtain an estimation of the treatment effect. This strategy is a heuristic extension of Kallus et al. (2018) to a non-linear case in the sense that the latent variables encode non-linear relationship between covariates.

An alternative, still heuristic, approach is to use the estimated latent confounders from (10) as inputs for standard techniques to estimate the average treatment effect. More precisely, for the doubly robust estimator (1), we replace the estimates for the propensity score with estimates for

$$\tilde{e}(z) = \mathbb{P}(W_i = 1 \mid \hat{Z}_i(x^\star) = z),$$

and similarly for the conditional response surfaces.

However, note that this latter strategy would require $\hat{Z}(x^\star)$ from (10) to be a confounder instead of $Z$ as it is assumed (see Figure 2).

## 4 SIMULATION STUDY

### 4.1 METHODS

We compare the following methods to handle missing values (the following acronyms are identical to the method labels used in Figures 3–5):

- MissDeepCausal:
  - `MDC.process`: using the doubly robust estimator MDC-process from Section 3.3;
  - `MDC.mi`: using the doubly robust estimator MDC-mi Section 3.2.

  We extended the publicly available code of Mattei & Frellsen (2019) to implement both methods, using the default parameters for the MIWAE part. Using notations of Appendix A, we use $L = 10,000$ for the importance sampling weights. In addition, for MDC-mi, we sample $B = 200$ observations from the estimated posterior distribution of $Z|X^\star$.

- `MICE`: the multiple imputation approach as suggested in Mattei & Mealli (2009) and Seaman & White (2014). We generate 10 imputations, using the implementation in the R (R Core Team, 2018) package `mice` (van Buuren & Groothuis-Oudshoorn, 2011).

- `MF`: the matrix factorization approach of Kallus et al. (2018) using R packages `softImpute` (Hastie & Mazumder, 2015) (for continuous data) and `mimi` (Robin, 2019) (for mixed data) for the matrix completion based on nuclear norm penalty. We choose the dimension of the latent space via cross-validation.

- `MIA`: the method proposed by Mayer et al. (2019b) which targets (4) and the generalized response surface analogue. It is based on estimation using random forests where missing values are encoded with *missing incorporated in attributes* such that the splitting rules in the random forests exploit the missingness pattern (Twala et al., 2008; Josse et al., 2019). We use the R package `grf` (Tibshirani et al., 2018) for the complete case and the implementation provided by Mayer et al. (2019b) for the incomplete case[1].

### 4.2 SETTINGS

Under the latent confounding assumption (corresponding to the graphical model in Figure 2), we generate covariates according to two models:

- LRMF: The covariates are generated from a low-rank matrix factorization model as in Kallus et al. (2018).

- DLVM: The covariates are generated from a deep latent variable model as in as in Kingma & Welling (2014). $Z_i \sim \mathcal{N}_d(0,1)$, covariates $X_i$ are sampled from $\mathcal{N}_p(\mu_{(Z)}, \Sigma_{(Z)})$, where $(\mu_{(Z)}, \Sigma_{(Z)}) = (V \tanh(UZ + a) + b, \text{diag}\{\exp(\eta^T \tanh(UZ + a) + \delta)\})$ with $U, V, a, b, \delta, \eta$ drawn from Standard Gaussian distributions and Uniform distributions.

Missing values are generated completely at random (MCAR), i.e., $\mathbb{P}(M_{ij} = 1) = \rho$, $\forall i$, $\forall j$, with $\rho \in \{0, 0.1, 0.3\}$ and we consider the following problem dimensions: $n = 10,000$, $p = 10$, and $d = 3$. Results are reported using 20 simulations for each setting. Throughout all experiments the true ATE $\tau$ is fixed at 1. Additional experiments with other choices of parameters are reported in the Appendix B.

### 4.3 RESULTS

#### 4.3.1 REGRESSION ADJUSTMENT

First, we assess the quality of our heuristic described in Section 3.3 concerning the non-linear extension of Kallus et al. (2018). For this we define treatment and outcome models with a logistic-linear model as follows: $\text{logit}(e(Z_{i.})) = \alpha^T Z_{i.}$ and $Y_i \sim \mathcal{N}((\beta^T Z_i + \tau W_i, \sigma^2)$. An estimation of $\tau$ is obtained by regressing the observed outcomes $Y_i$ on the estimations of the latent factors $Z$ (for `MDC.process`, `MF`) and on the imputed data $X_{imp}$ (for `MICE`).

---

[1]https://github.com/imkemayer/causal-inference-missing

Figure 3 shows that as expected our proposed method, `lin.MDC.process` actually outperforms all other methods when the covariates are generated according to a DLVM model. The bias observed is small with respect to the one exhibited for completely observed covariates $X$. Additionally we observe that if the data is generated under the LRMF model, then our method performs as well as the initial proposal of Kallus et al. (2018) (results for this are not reported here).

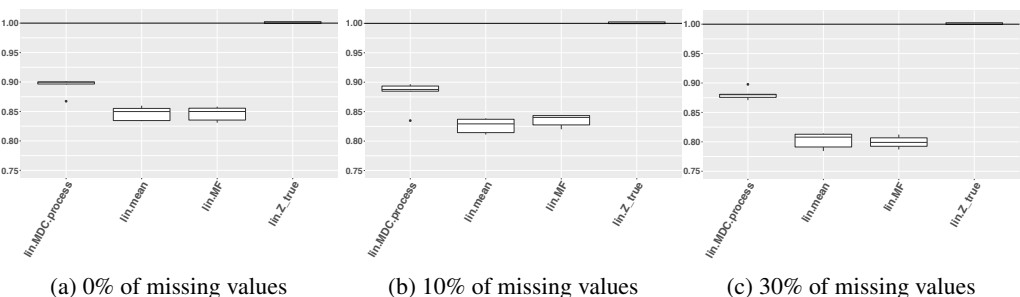

(a) 0% of missing values      (b) 10% of missing values      (c) 30% of missing values

Figure 3: Estimated ATE via regression adjustment; covariates generated from a DLVM, (logistic-)linear model specification for $(e, \mu_0, \mu_1)$; results with `Z_true` results are obtained using the true confounders $Z$.

### 4.3.2 DOUBLY ROBUST ESTIMATION

Now we turn to the more flexible framework which does not assume linear relationships (equation 5) between the outcome and the confounders. We generate the treatment and outcome using logistic-linear models and non-linear models based on non-linear transformations of the latent variables $Z$. We consider the doubly robust estimator (1) with the (imputed) covariates $X$ for MICE and with the estimation of the latent variables $Z$ for MF and MissDeepCausal.

To estimate the regression surfaces $(\mu_1, \mu_0)$ and the propensity score $e$ required for the doubly robust estimator (equation 1), we use either a logistic-linear model or (generalized) random forests (Athey et al., 2019), indicated respectively by the prefixes `loglin` and `grf` in all figures.[2] For the latter, we use the implementation of the R package `grf` (Tibshirani et al., 2018).

Figure 4 illustrates that even when the latent variables are generated from matrix factorization, our approaches based on the VAE with missing values lead to estimates that are almost unbiased, given that the estimation of the propensity score and response surfaces is adapted to the considered models. The small bias observed for the matrix factorization pre-processing approach from Kallus et al. (2018) is not in contradiction with their theory since we use the doubly robust estimators and not the regression model. In addition, they require a (much) larger number of proxy covariates w.r.t. the number of latent confounders.

Figure 5 shows that as expected, due to the flexibility of MissDeepCausal, the suggested approaches better handle highly non-linear relationships between the latent confounders and the observed (incomplete) covariates. It turns out that the multiple imputation strategy is particularly appropriate when the relationships between the outcome, the treatment and the confounders are highly non-linear.

### 4.4 IHDP DATA

We assess our methodology on the Infant Health and Development Program (IHDP) benchmark data (Hill, 2011). The original data comes from a randomized control trial where the aim was to assess the impact of visits by specialists on children's test scores. There are six quantitative and 19 binary variables, recorded for 985 individuals. Hill (2011) transformed the original experimental data into observational data by selecting a nonrandom subset among the treated, stratified along an ethnicity variable, which leads to two unbalanced treatment groups. In total there are 139 treated and 608 control observations in the new data set. Then, keeping fixed the treatment variable, simulated

---

[2]More specifically, for estimating $\mu_w$, we use all observations having $W_i = w$.

[2]In Figure 4b, all `loglin` estimations yield values around 6 and are therefore omitted for better readability.

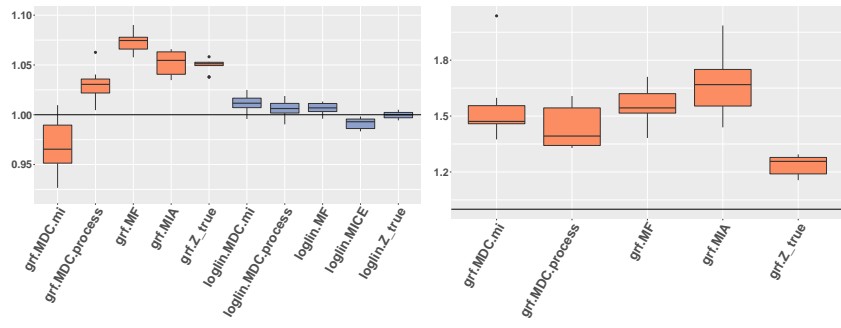

(a) Logistic-linear model specification for $(e, \mu_0, \mu_1)$

(b) Non-linear model specification for $(e, \mu_0, \mu_1)$

Figure 4: Estimated ATE for 10% of missing values; covariates are generated according to LRMF; Z_true results are obtained using the true confounders $Z$.

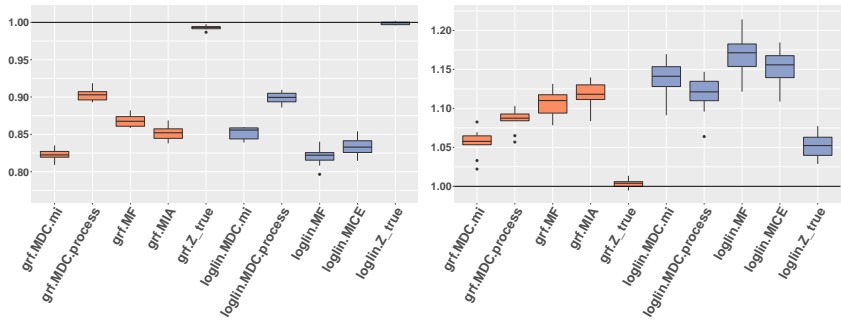

(a) Logistic-linear model specification for $(e, \mu_0, \mu_1)$

(b) Non-linear model specification for $(e, \mu_0, \mu_1)$

Figure 5: Estimated ATE for 10% of missing values; covariates are generated according to DLVM; Z_true results are obtained using the true confounders $Z$.

data are obtained by generating new potential outcomes. More precisely, we follow the scenario "B" of Hill (2011) , i.e., $Y(0) \sim \mathcal{N}(\mu_0, 1)$ and $Y(1) \sim \mathcal{N}(\mu_1, 1)$, with $(\mu_0, \mu_1) = (\exp(X + W)\beta, X\beta - \omega)$ where $\omega$ is chosen to get an average treatment effect $\tau$ equal to 4. [3] After simulating the outcomes, we add missing values to the 25 covariates, assuming an MCAR mechanism. For the MIWAE part of our MDC methods, we select the parameters $\sigma_{prior}$ and $d$ by 5-fold cross-validation.

In addition to comparing the estimators considered in this paper that handle missing data, we also add the CEVAE estimator detailed in Louizos et al. (2017) as a baseline. Note that CEVAE does not deal with missing values so that we replace the missing values by the mean of the variables. The CEVAE estimator is based on the difference between the two conditional expectations. For comparability with their method and previous experiments on these data, we report the in-sample mean absolute error, i.e. the mean absolute difference between the estimated ATE and the sample ATE (by construction of the data we know the exact values of $\mu_{(1)}(X_i)$ and $\mu_{(0)}(X_i)$ for all $i$): $\Delta = \left| \hat{\tau} - \frac{1}{n} \sum \mu_{(1)}(X_i) - \mu_{(0)}(X_i) \right|$.

Table 1 shows that the doubly robust estimators (either in the parametric regression, $DR_{log-lin}$, or the random forest form $DR_{rf}$) systematically outperform the corresponding OLS estimator which highlights that the linear model is not appropriate, at least that it is not linear in the covariates $X$. Indeed, we know that the outcome is simulated as a nonlinear function of the (complete) covariates $X$, whereas the treatment assignment is taken from the (derandomized) experiment and can therefore well depend on latent variables. The results of MissDeepCausal are competitive with other

---

[3]We use and adapt the corresponding code from V. Dorie: `https://github.com/vdorie/npci/`.

| % NA | Method | Δ | | |
|------|--------|-----|--------------|-----------|
| | | OLS | $DR_{log-lin}$ | $DR_{rf}$ |
| 0 | $X$ (complete data) | $0.72 \pm 0.02$ | $0.13 \pm 0.00$ | $0.20 \pm 0.01$ |
| | $MF$ | $0.56 \pm 0.03$ | $0.14 \pm 0.01$ | $0.16 \pm 0.01$ |
| | $MDC.process$ | $0.51 \pm 0.03$ | $0.15 \pm 0.01$ | $0.19 \pm 0.03$ |
| | $MDC.mi$ | $0.47 \pm 0.03$ | $0.16 \pm 0.01$ | $0.14 \pm 0.02$ |
| | $CEVAE(X)$ | | $0.34 \pm 0.02$ | |
| 10 | $MICE$ | $0.85 \pm 0.02$ | $0.16 \pm 0.00$ | $0.24 \pm 0.01$ |
| | $MIA.GRF$ | – | – | $0.23 \pm 0.01$ |
| | $MF$ | $0.50 \pm 0.03$ | $0.15 \pm 0.01$ | $0.15 \pm 0.01$ |
| | $MDC.process$ | $0.42 \pm 0.02$ | $0.15 \pm 0.01$ | $0.16 \pm 0.02$ |
| | $MDC.mi$ | $0.35 \pm 0.02$ | $0.17 \pm 0.01$ | $0.13 \pm 0.02$ |
| | $CEVAE(X_{mean\_imp})$ | | $0.31 \pm 0.01$ | |
| 30 | $MICE$ | $1.20 \pm 0.02$ | $0.30 \pm 0.00$ | $0.32 \pm 0.01$ |
| | $MIA.GRF$ | – | – | $0.17 \pm 0.01$ |
| | $MF$ | $0.39 \pm 0.02$ | $0.16 \pm 0.01$ | $0.17 \pm 0.01$ |
| | $MDC.process$ | $0.37 \pm 0.02$ | $0.16 \pm 0.01$ | $0.15 \pm 0.02$ |
| | $MDC.mi$ | $0.30 \pm 0.02$ | $0.18 \pm 0.01$ | $0.13 \pm 0.01$ |
| | $CEVAE(X_{mean\_imp})$ | | $0.38 \pm 0.02$ | |
| 50 | $MICE$ | $1.54 \pm 0.03$ | $0.46 \pm 0.01$ | $0.42 \pm 0.01$ |
| | $MIA.GRF$ | – | – | $0.19 \pm 0.01$ |
| | $MF$ | $0.28 \pm 0.01$ | $0.20 \pm 0.01$ | $0.21 \pm 0.02$ |
| | $MDC.process$ | $0.24 \pm 0.01$ | $0.20 \pm 0.01$ | $0.21 \pm 0.02$ |
| | $MDC.mi$ | $0.18 \pm 0.01$ | $0.22 \pm 0.01$ | $0.22 \pm 0.03$ |
| | $CEVAE(X_{mean\_imp})$ | | $0.38 \pm 0.02$ | |

Table 1: Methods on the IHDP benchmark data. Mean absolute error $\Delta$ (with standard error) across simulations on all the data points (in-sample error). OLS corresponds to the estimator obtained by regression and DR to the doubly robust estimator(s).

approaches and greatly improve on CEVAE and MICE. Its performances when used with the double robust estimators are stable with respect to the percentage of missing values.

## 5 CONCLUSION

In this work we have investigated the problem of treatment effect estimation with incomplete co-variates. This problem of missing values is highly relevant for modern causal inference as it is exacerbated with high dimensional data. Yet most causal inference techniques do not address this issue; and complete case analysis, in addition to leading to potentially inconsistent causal effects estimators, is not an option anymore. We have proposed MissDeepCausal which borrows the strength of deep latent variable models to retrieve the latent confounders from incomplete covariates encoding complex non-linear relationships. We use a modular approach in the style of Bayesian propensity based methods for treatment effect estimation (Zigler, 2016), where the latent variables are used as inputs for doubly robust estimators. We suggest a multiple imputation strategy that allows to fully exploit the posterior distribution of the latent variables. Numerical results are very encouraging insofar as we obtain best relative performance in terms of bias whether the underlying model is well or badly specified compared to current state of the art. Open challenges include heterogeneous treatment effect estimation with missing values as well as the ambitious task of handling missing not at random type data.

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

## A   APPENDIX

### A.1   DEEP LATENT VARIABLE MODELS WITH MISSING VALUES

Deep latent variable models can be defined as follows. Let $(X_i,\ Z_i)_{i \leq n}$ be $n$ i.i.d. random variables such that

$$\begin{cases} Z_i \sim P(Z_i) \\ X_i \sim P_\theta(X_i|Z_i) = \Phi\left(X_i|f_\theta(Z_i)\right). \end{cases}$$

The prior distribution of the latent variables or *codes* $Z_i \in \mathbb{R}^d$ is often isotropic Gaussian $Z_i \sim \mathcal{N}(0_d, I_d)$. The function $f_\theta : \mathbb{R}^d \to H$ is a (deep) neural network called the *decoder* and $\Phi(\cdot|\eta)_{\eta \in H}$ is a parametric observation model, which we take to be multivariate Gaussian. The inference of deep latent variable models can be achieved by maximizing evidence lower bounds of the likelihood, such as the variational autoencoder bounds.

With missing values, the appropriate quantity to target for inference on $\theta$, when the missing values mechanism can be ignored (Rubin, 1976; Little & Rubin, 2002), is the observed log-likelihood. Using Rubin (1976)'s notations, we define $X_i = (X_{i,obs}, X_{i,mis})$ the partition of the data in realized observed and missing values given a specific realization of the pattern, it can be written as:

$$\ell(\theta) \triangleq \sum_{i=1}^{n} \log p_\theta\left(X_{i,obs}\right) = \sum_{i=1}^{n} \log \int p_\theta\left(X_{i,obs}|Z_i\right) p(Z_i) dZ_i.$$

The corresponding evidence lower bound (ELBO) is:

$$\mathcal{L}(\theta, \gamma) \triangleq \sum_{i=1}^{n} \mathbb{E}_{Q_\gamma}\left[\ln P_\theta\left(X_{i,obs}|Z_i\right)\right] - KL\left(Q_\gamma\left(Z_i|X_{i,obs}\right)\|P_\theta\left(Z_i\right)\right),$$

with $KL$ for the Kullback-Leibler divergence and the variational distribution

$$Q_\gamma\left(Z|X_{obs}\right) \triangleq \Psi\left(Z|g_\gamma(X_{obs})\right),$$

with $\Psi(\cdot)$ the (parametric) variational distribution over $\mathbb{R}^d$. The function $g_\gamma : \mathcal{X} \to \mathcal{K}$, called the *encoder*, is parametrized by a (deep) neural network whose weights are stored in $\gamma \in \Gamma$.

To take into account missing values in deep latent variable models, Mattei & Frellsen (2019) suggest the missing data importance weight autoencoder bound (MIWAE) approach. They use a simple variational family where they impute the missing entries with a constant and show that using this class of distributions, it maximizes a lower bound of the observed log-likelihood. Specifically, they replace $Q_\gamma$ with

$$Q_\gamma\left(Z|X_{obs}\right) = \Psi\left(Z|g_\gamma\left(\iota\left(X_{obs}\right)\right)\right),$$

where $\iota$ is an imputation function chosen beforehand that transforms $X_{obs}$ into a complete input vector $\iota\left(X_{obs}\right) \in \mathcal{X}$.

## B    Supplementary results

We only report the cases with logistic-linear specification of $(e, \mu_0, \mu_1)$, we therefore use the logistic-linear doubly robust estimator and omit the prefix loglin in Figures 6 and 7. They show that when the number of covariates is large the MDC methods accurately recover the true ATE. In the DLVM case (bottom row), the MDC methods outperform all other compared methods.

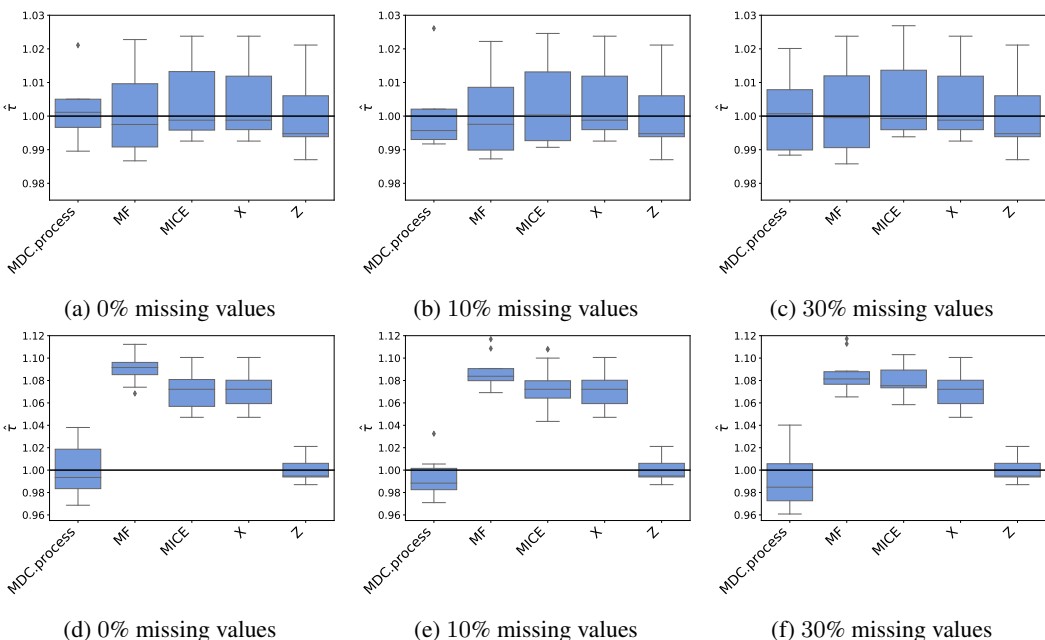

(a) 0% missing values  (b) 10% missing values  (c) 30% missing values

(d) 0% missing values  (e) 10% missing values  (f) 30% missing values

Figure 6: Estimated ATE for different amounts of missing values with **regression estimation**; $p = 100$ covariates, $n = 1,000$ observations, $d = 3$ latent confounders, logistic-linear model specification for $(e, \mu_0, \mu_1)$; top: LRMF; bottom: DLVM.

Additionally, we report results on the effect of changing the sampling parameter $B$ for the MDC.mi strategy (all other parameters of the VAE are set to the default values). Again, we only report the cases with logistic-linear specification of $(e, \mu_0, \mu_1)$. These results, in Figure 8, show that for small number of covariates (in this case, $p = 5$), the choice of $B$ does not have a large influence on the final estimate, however for larger number of covariates (e.g., $p = 100$), increasing $B$ reduces the bias and the variance.

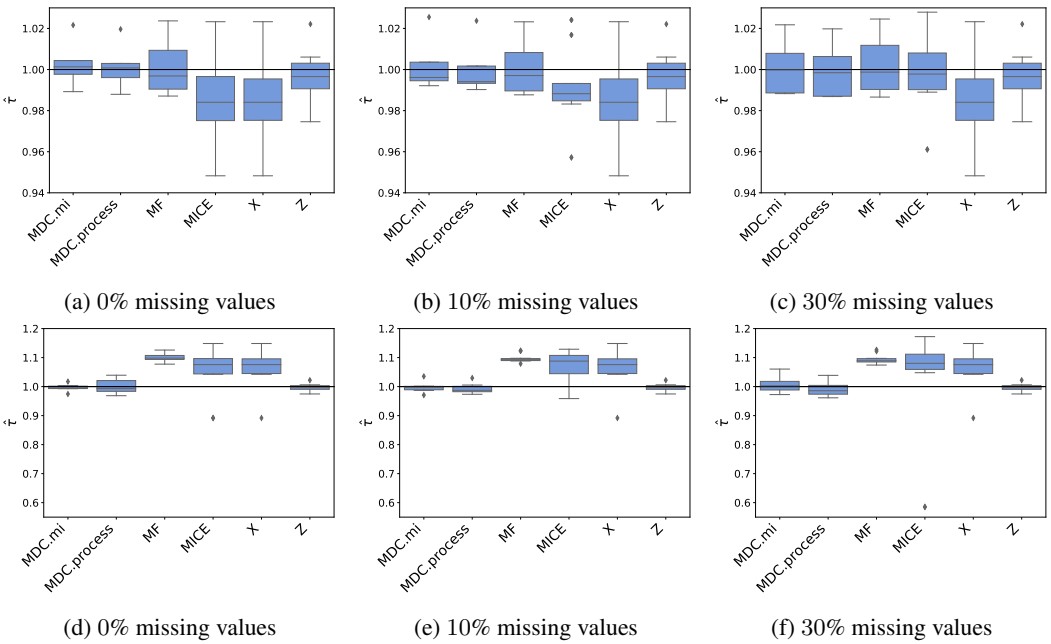

Figure 7: Estimated ATE for different amounts of missing values with **doubly robust estimation**; $p = 100$ covariates, $n = 1,000$ observations, $d = 3$ latent confounders, logistic-linear model specification for $(e, \mu_0, \mu_1)$; top: LRMF; bottom: DLVM.

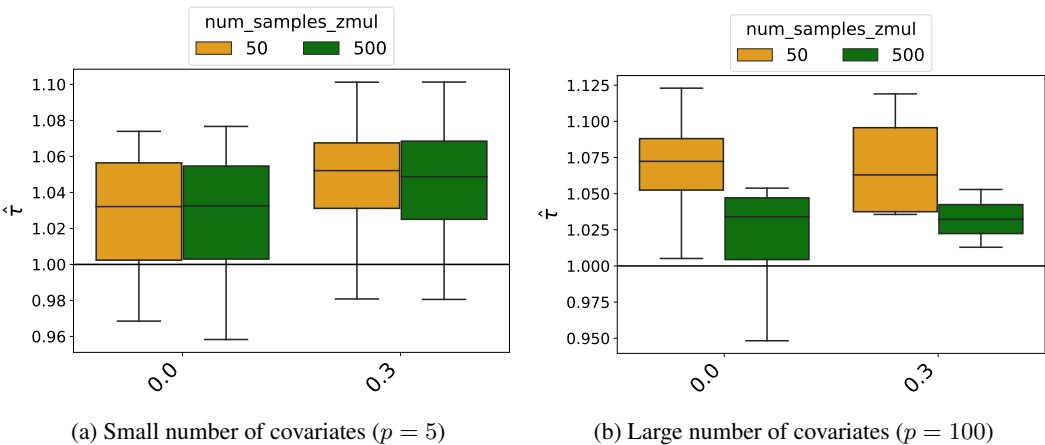

Figure 8: Estimated ATE with MDC.mi + **doubly robust estimation** for different choices of $B$; DLVM setting, $n = 1,000$ observations, $d = 3$ latent confounders, logistic-linear model specification for $(e, \mu_0, \mu_1)$.

