# OpenReview forum: "MissDeepCausal: causal inference from incomplete data using deep latent variable models"
_ICLR.cc/2020/Conference — Reject_

### Official Review · AnonReviewer3 · 2019-10-23
**Official Blind Review #3**

**Rating:** 6

**Review:**

This paper introduces MissDeepCausal method to address the problem of treatment effect estimation with incomplete covariates matrix (missing values at random -- MAR). It makes use of Variational AutoEncoders (VAE) to learn the latent confounders from incomplete covariates. This also helps encoding complex non-linear relationships in the data, a capability that is missing in the work of Kallus et al. (2018) -- the work which this paper extends. They employ the Missing data Importance Weight AutoEncoder (MIWAE) approach (Mattei & Frellsen, 2019) to approximate the posterior of their latent factors Z given the observed incomplete covariates X*. The main contributions of this work are presented in sections 3.2 and 3.3, where they use the approximated posterior derived from MIWAE to sample Z to be used for estimating outcomes and finally calculating the Average Treatment Effect (ATE). This is done according to the doubly robust estimator developed for data with incomplete covariates (Mayer et al., 2019b).

In summary, I am not convinced that the contribution of this paper is enough, nor of its novelty. However, I will read the rebuttal carefully and am willing to increase the score if the authors address this concern.

There are several points that need further clarification; e.g.,
	- Figure 1 as well as Figure 2 show a directed edge from X* to X_{miss}. Does this mean that X* has all the proxies needed to identify X_{miss}?
	- How does this method assure/evaluate that Z embeds enough information to predict accurate effects?
	- How are \mu_0 and \mu_1 functions trained on Z

Things to improve the paper that did not impact the score:
	- Page 2, par. 2, last line: state-of-the-art method”s”
	- Page 3, under Unconfoundedness par., line -7: [...] for each observation “comma” treatment assignment [...]
	- Page 3, Figure 1: According to ICLR’s formatting guidelines, the figure number and caption must always appear after the figure.
	- Page 3, Missingness par., line 1: [...] is one “of” the most [...]
	- Page 5, line after Eq. (8): 8 should be in parentheses.
	- Page 7, Figure 3: box-plots are hardly legible.
	- Page 7, Figure 3 caption, line 2: keep “(logistic-)linear” together with \mbox{} or ~ in latex

References:
	- Kallus, N., Mao, X., & Udell, M. (2018). Causal inference with noisy and missing covariates via matrix factorization. In Advances in neural information processing systems (pp. 6921-6932).
	- Mattei, P. A., & Frellsen, J. (2019). MIWAE: Deep Generative Modelling and Imputation of Incomplete Data Sets. In International Conference on Machine Learning (pp. 4413-4423).
	- Mayer, I., Wager, S., Gauss, T., Moyer, J. D., & Josse, J. (2019). Doubly robust treatment effect estimation with missing attributes. preprint.


********UPDATE after reading the rebuttal********
The authors have provided further clarifications in their rebuttal and therefore, I increased my score form “weak reject” to “weak accept”.



**Experience Assessment:**

I have published one or two papers in this area.

**Review Assessment: Checking Correctness Of Derivations And Theory:**

I assessed the sensibility of the derivations and theory.

**Review Assessment: Checking Correctness Of Experiments:**

I assessed the sensibility of the experiments.

**Review Assessment: Thoroughness In Paper Reading:**

I read the paper thoroughly.

---

> ### Author Response · Authors · 2019-11-08
> **Revised version better highlights the contribution**
>
> Thank you for your comments.
>
> - Regarding the novelty and contribution of the paper: this is of course a subjective judgment, but we provide a detailed description of the state-of-the-art in the paper to clarify the original contribution of the paper, namely, the first model to estimate ATE with latent confounding variables and missing values in observed features, without assuming linear relationships among variables.
> In the revised version of the paper we added in particular a paragraph in Section 3 page 4 to clarify that the MDC.mi method we propose provides an asymptotically unbiased ATE estimation if the VAE correctly estimates the conditional distribution of the latent factors given the observations. To strengthen the paper and highlight our contribution with multiple imputation, we have also included in the Appendix of the revised version of the paper the results of other simulations where we vary the number of covariates that highlight the fact that we get unbiased estimates of the ATE when $p$ increases. Finally, we would like to stress out that there are not so many methods that explicitly consider missing (covariate) values in causal inference tasks. The assumptions on the missing values mechanism (MCAR for Kallus et al., 2018) or on the unconfoundedness (unconfoundedness with missingness, Mayer et al., 2019) are strong and it is difficult to assess their suitability for a given problem. Therefore we believe that  there is room for our proposal which bears an innovative approach for exploiting the latent confounding assumption in the sense that using the multiple imputation approach, we exploit the posterior distribution of $(Z|X^*)$ instead of only the posterior expectation $\mathbb{E}[Z|X^*]$.
>
> - Regarding Figures 1 and 2: We agree that the directed edge from $X^*$ to $X_{mis}$ is confusing. It stems from an alternative representation where we used $X_{mis}$ and $X_{obs}$, which is classical in the missing data literature. Nevertheless, we agree that the assumptions are clearer without any $X_{mis}$ in the graphical representation but only $X^{\star}$ and $M$. Consequently, we have modified the graphs in Figure 1 and Figure 2 of the revised version and added also $Y(0)$ and $Y(1)$ to highlight the unconfoundedness assumption.
>
> - About the validity of using Z to predict ATE accurately: Like all methods for ATE estimation, MDC is only valid under some assumptions that we state in the paper, in particular, that there are no other confounding factors between treatment and effects beyond the (unobserved) Z factors (i.e., that the model shown on Figure 2 is correct). Ensuring and evaluating the validity of this hypothesis is unfortunately usually not possible without further assumptions, as for other methods for ATE estimation. If the model is correct, then we added a paragraph (Section 3 page 4) to justify by the MDC.mi estimator is a valid estimator for ATE.
>
> - About the training of the conditional response surfaces $\mu_0$ and $\mu_1$: As explained in Section 4.3.2, we estimate $\mu_0$ and $\mu_1$ by performing a predictive model of $Y$ on $Z$ using either a linear regression or a random forest.
>
> - We have corrected all the typos, thank you.

---

### Official Review · AnonReviewer2 · 2019-10-24
**Official Blind Review #2**

**Rating:** 6

**Review:**

This contribution considers deep latent-factor models for causal inference -inferring the effect of a treatment- in the presence of missing values. The core challenge is that of confounders not directly observed, and accessible only via noisy proxys, in particular in the missingness. The contributed method relies on using the latent-factor model for multiple imputation in doubly-robust causal treatment effect estimators. As a consequence, it requires the missing at random assumption to control for the impact of imputation. Given that the confounders are not directly assumed, a first approach estimates their effect via an estimate of P(Z|X*)  (probability of confounder given observed data), which is then plugged in the doubly robust estimator in a multiple imputation strategy. A second approach uses heuristically the estimated latent confounders as regressors of non interest in a linear-regression model. The approaches are empirically compared to other imputation strategies used as plugins in the doubly-robust estimator. The contributed approach show marked benefits when the problem is highly non-linear.

The manuscript is clearly written.

I do not have many comments.

One concern though is that the VAE comes with a significant amount of hyper-parameters that do not seem obvious to set. This is to be contrasted with other approaches compared to. How was the specific architecture and learning strategy of the VAE selected?

The simulation settings are somewhat artificial. More simulations inspired from real-life causal scenario would improve the work.

I hope that in the final version, the code will be available publicly, and not on request.

**Experience Assessment:**

I have published one or two papers in this area.

**Review Assessment: Checking Correctness Of Derivations And Theory:**

I assessed the sensibility of the derivations and theory.

**Review Assessment: Checking Correctness Of Experiments:**

I assessed the sensibility of the experiments.

**Review Assessment: Thoroughness In Paper Reading:**

I read the paper thoroughly.

---

> ### Author Response · Authors · 2019-11-08
> **Comments taken into account**
>
> Thank you for  your positive feedback.
>
>  - We forgot to indicate in the initial version that we used the default parameters for the VAE as proposed in the implementation of the authors of  MIWAE available at \url{https://github.com/pamattei/miwae}. We have now added this information in the updated version of our article.
>
> - We included results of new simulations where we vary the number of observed covariates in the revised version of the paper (Figs 6-7). It allows us to highlight even more the good performances of our methods as it leads to unbiased estimates when $p$ increases.
>
> - The code is not public yet just to keep the review process double-blind, but the code will be available on GitHub with all scripts needed to reproduce all experiments, which is crucial for reproducibility of our work. We remove the sentence mentioning the availability of the code "upon request" in the revised version of the paper.

---

### Official Review · AnonReviewer1 · 2019-11-06
**Official Blind Review #1**

**Rating:** 6

**Review:**

Summary:
       The paper considers average treatment effect estimation treatment T and an unobserved confounder Z causes the outcome Y with an added constraint that the observed X is a noisy measurement of the underlying Z and some of the entries of observed X are missing at random (MAR). Previous work (Kallus et al. 2018) on similar settings assumed a low rank model connecting Z and X along with some entries missing at random which we do not observe. Further, Y (outcome) is related to the treatment and Z with a linear model. They actually show that matrix factorization techniques with these assumptions form an unbiased estimator for the Average Treatment Effect. There is prior work on doubly robust estimators under ignorability assumptions.

In this paper, the authors want to consider general non-linear relationships between Z and X with the same MAR (missing at random assumption) for missing entries.  So they fit a latent variable model in the form of VAE to find the P(Z| observed part of X) using a slightly modified version of the ELBO Lower Bound. For missing entries, they just replace those entries by a constant and do the usual VAE fit. After the VAE fit, multiple Z's are samples from the optimized P(Z| observed X) and then used in the doubly robust formula on each Z for estimating the average treatment effect and then finally the estimates averaged over the different Z's.

There is an alternative where the conditional mean of the latent variable is estimated from the VAE and used in the doubly robust computation.

In many synthetic examples, authors compare this with existing baselines and show that their method improves.

Pros:
 - Baselines compares are comprehensive enough from my perspective.
 - The paper is well written with clear pointer to existing work on doubly robust estimators with standard ignorability assumptions and the work for the linear, low rank model case by Kallus et al. 2018.

Cons:
   Major Issues
     - There is no reason to believe that even for the synthetic experiments, that the VAE posteriors would asymptotically yield unbiased ATE's which is provably the case in (Kallus et al. 2018) (of course for their restricted linear model/low rank assumptions). There is no reason to suppose Z's used in eq (9) from the VAE satisfy ignorability even in the asymptotic limit. In this light, the paper in essence just estimates Z's from some latent model that is fit and then use those latents to regress Y and then computes ATE. This seems a natural heuristic to try given such a problem. So I don't find a big methodological novelty. I would be willing to increase my scores if the authors could convince me on this point.

  - For the LRMF model (Fig 4) the MF approach seems to do as well as the authors proposal and we have a guarantee for the MF case under those linear/low rank modeling assumptions. So the only demonstrated benefit is for the synthetic experiments for Fig 5 and Fig 3 (I agree that it is considerable particularly with large fraction of missing values in Fig 3) in whose settings we dont know about how unbiased it is in the limit. More synthetic experiments with different kind of generative models could be more convincing.

- Some real world data set would have been more convincing - although I agree ground truth is hard to come by.

Minor Issues:
   - You have set B=200 for all the experiments for MDC-MI. Do the results change when B is increased or decreased ?? Does variance go down or the bias itself changes with B -  This would be a useful insight to have.

  -  There is typo in the ELBO lower bound equation in page 11. There are other minor typos. Please correct for it.

 - Since the paper is about estimate treatment effect from measurements of an unobserved confounder - it is important to cite - https://ftp.cs.ucla.edu/pub/stat_ser/r366-reprint.pdf from the causal DAG literature.

 - The covariance of X given Z for the DLVM model is not clear - It seems to say exp ( or some matrix vector products) * Identity. What does this mean ?

- The feature dimensions seems to be set at 10 - so would we expect the same results in much higher dimensions - like say 100s for few tens of thousands of samples??

********UPDATE after reading the rebutall,changes and the new experiment**********

I appreciate the authors actually accepting that identifiability issues cannot be easily resolved even if one knows P*(Z|X).
I recommend the authors to elaborate on this point in the camera ready version. However, showing that the proposed methods work on a real benchmark (semi-synthetic one used in Shalit et. al 2017) is commendable.  However, I find that the MF method is competitive (almost all the time) with their method when Doubly robust estimators are used.

But having matched an existing baseline (the MF method) that deals with confounders on real data and showing superior synthetic results and authors clarifying and toning down their theoretical claims, I am inclined to increase the score to Weak accept.







**Experience Assessment:**

I have published one or two papers in this area.

**Review Assessment: Checking Correctness Of Derivations And Theory:**

I assessed the sensibility of the derivations and theory.

**Review Assessment: Checking Correctness Of Experiments:**

I assessed the sensibility of the experiments.

**Review Assessment: Thoroughness In Paper Reading:**

I read the paper thoroughly.

---

> ### Author Response · Authors · 2019-11-08
> **Multiple imputation strategy asymptotically unbiased for ATE**
>
>
> We would like to thank the reviewer for their comments and careful reading of the paper.
>
> - About the methodological novelty and the properties of our ATE estimator: while we have no theoretical claim about the MDC.process estimator, which we propose merely as a natural nonlinear generalization of the matrix factorization technique of Kallus et al (2018), we would like to clarify that the MDC.mi estimator is asymptotically unbiased for ATE under the latent confounding assumption, if we assume that (i) the VAE asymptotically estimates the correct conditional distribution of the latent variable given the observed features, and (ii) we use an asymptotically consistent estimator of ATE given the latent variables, such as the double robust estimator (Chernozhukov et al., 2018). We clarify this original methodological contribution in the revised version of the paper, where we added a paragraph in Section 3 page 4 to state and prove this property. Of course, a complete answer to the question of asymptotic consistency would require us to answer point (i), i.e., study the asymptotic consistency of VAE and quantify how well we approximate the conditional distribution of the latent variables given the observed data when we sample with MIWAE, which is beyond the scope of this paper.
>
> - Concerning the experiments: First, we would like to clarify that the experiment with LRMF model (Fig 4) mentioned by the Reviewer is precisely meant to give an advantage to the MF approach, since the simulated data follow precisely the model behind the MF method; hence it is not a surprise that the MF approach is competitive in this case. To the contrary, as we write in the text, the interesting observation in this experiment is that MDC approaches are competitive with MF, and even slightly better. Second, we agree that more experiments are needed to evaluate empirically the asymptotic biases of the different methods. In the revised version, we add new results (figures 6 and 7 in appendix B) showing that MDC outperforms other methods in terms of bias as the number of features increases. For instance, choosing $p=100$ and $n=1000$, we obtain unbiased estimations of the ATE with both MDC.process and MDC.mi coupled with the double robust estimator in both the low-rank and deep-latent variables settings. Regarding real world data, as noticed by the reviewer ground truth is hard to come by.
> - Regarding the influence of $B$, we only have preliminary results but these indicate that for a small number of covariates, choosing small ($B=50$) or large ($B=500$) values for $B$ does not have an impact on the final estimate, however for larger numbers of covariates, it seems better to pick a large $B$. We have added these results in Figure 8 in appendix B of the revised paper.
>
> - We have corrected the typos and added the reference, thank you.
>
> -  Regarding the covariance of X given Z for the DLVM model: we use the model suggested by Kingma and Welling (2014) who used a diagonal covariance matrix, i.e., conditionally on $Z$, we have independence between the (normally distributed) covariates $X$. Hence we could also  write $(\mu_{(Z)},\Sigma_{(Z)}) = (V\tanh(UZ+a)+b, \operatorname{diag}\{\exp(\eta^T\tanh(UZ+a) + \delta\})$
>
> - Regarding the feature dimension, as explained above we added in the revised version new results with $p=100$ and $n=1000$ where we show that MDC is unbiased (Figs 6-7)
>
>
> References:
> - Kallus, N., Mao, X., \& Udell, M. (2018). Causal inference with noisy and missing covariates via matrix factorization. In \textit{Advances in neural information processing systems} (pp. 6921--6932).
> - Diederik P. Kingma and Max Welling. Auto-encoding variational bayes. In \textit{International Conference on Learning Representations}, 2014.
> - Victor Chernozhukov, Denis Chetverikov, Mert Demirer, Esther Duflo, Christian Hansen, Whitney Newey, and James Robins.   Double/debiased machine learning for treatment and structural parameters. \textit{The Econometrics Journal}, 21(1):C1--C68, 2018.

---

> > ### Comment · AnonReviewer1 · 2019-11-08
> > **Synthetic experiments with more covariates seems promising but arguments about unbiasedness not convincing**
> >
> > I thank the authors for taking time to update the manuscript considering my comments previously. I have some concerns still.
> >
> > "We could also write $(\mu_{(Z)},\Sigma_{(Z)}) = (V\tanh(UZ+a)+b, \operatorname{diag}\{\exp(\eta^T\tanh(UZ+a) + \delta\})$" - I would prefer it to be written this way.
> >
> > "Regarding the feature dimension, as explained above we added in the revised version new results with  and  where we show that MDC is unbiased (Figs 6-7)"
> > I agree that the results do indeed show that in higher dimensions this works better than even the initial reported results.
> >
> > (Most important concern still) "Of course, a complete answer to the question of asymptotic consistency would require us to answer point (i), i.e., study the asymptotic consistency of VAE and quantify how well we approximate the conditional distribution of the latent variables given the observed data when we sample with MIWAE, which is beyond the scope of this paper."
> > So the authors claim that if the correct P (Z|X*) is captured, then their estimates would be asymptotically unbiased due to the use of doubly robust estimators. However, I have the following problem with this claim. May be I am not able to see it clearly.
> >
> > Suppose (to make my point clear and in a simple way) there are only two data points Y_1,X*_1,T_1 and Y_2,X*_2,T_2. Let us assume the *true* latent *realizations* behind these points were *Z_1* and *Z_2* - lets say we don't observe Z_1 and Z_2 directly. However, let us say we also know P (Z|X*) distribution exactly. Now my question is  - Is the estimator you propose even unbiased (even asymptotically) ?
> >
> > Now when you sample B times from P(Z|X*) and create B tables - there is no reason that in each of the B tables, you will get points Z that are even close to Z_1 or Z_2 most of the times while the targets Y's correspond to these specific unknown Z_1 and Z_2. So if you form your regression estimates along with the doubly robust estimator based on the Z's one generates - there is no reason to believe it is unbiased. In fact, let us even take a stronger case where we know the treatment assignment conditional also P (T|Z) exactly. But the doubly robust estimates use regressed mean of Y (obtained only under Z_1) on Z's (sampled) given X* under T=T_1. Why is this unbiased even asymptotically ? The point I am making is we only know Y according to one Z_1 (sampled from P (Z|X*) and unobserved). Regressing this target on Z's that are sampled from the distribution (P(Z|X*)) - will it not make it biased - why would it be unbiased?
> >
> >  Is there a reference where this issue has been dealt which the authors are relying on ?
> >
> >  Do you think you can get close to estimates formed by the unobserved realizations (Z_1 and Z_2) with a regression estimate based on Z's samples from P (Z|X*)??
> >
> > - The only idenitifability result I know when Z is unknown in general comes from https://ftp.cs.ucla.edu/pub/stat_ser/r366-reprint.pdf (Kuroki and Pearl 2014). In fact they want to know P (X*|Z) and not P (Z|X*) [which might be counter intuitive] and they establish identifiability results. Again this is possibly through a different estimator since for them both Z and X* are discrete and there are no missingness issues. But to claim that P(Z|X*) alone is enough to identify through sampling with doubly robust estimator requires a proof.
> >
> > Regarding experiments:
> >     I am happy that the authors did synthetic experiments with higher p and it shows better results.
> >
> > Shalit et al 2017 used two real world datasets (one sort of semi synthetic) in https://arxiv.org/pdf/1606.03976.pdf for their ITE work. Is it possible to hide some variables and introduce missingness to create the situation the authors have and then perform an experiment using their data ? Anything close to real world data would add value to the experimental claims given that I am still not convinced by the identifiability claims that seem strong even knowing P(Z|X*) (in Section 3)

---

> > > ### Author Response · Authors · 2019-11-13
> > > **Clarification of the arguments and new simulations on benchmark data**
> > >
> > > We would like to thank the reviewer for his detailed and insightful comments.
> > >
> > > - Regarding the identifiability issue: we realize our arguments were not clear enough and have created some confusion, and are sorry for that.
> > >
> > > On the one hand, we fully agree with the reviewer that without further assumptions, the causal effect is in general not identifiable from the knowledge of $P(Z|X^*)$ only (plus an infinite number of observations of (W,Y,X*)). For example, if $X$ is independent from $Z$ (corresponding to a case where the observation is a very bad proxy for the latent confounder), then $P(Z|X)=P(Z)$ is perfectly known but the causal effect can not be estimated from the observation of $(W,Y)$ only and some noise $X^*$.
> > >
> > > On the other hand, what we wanted to show with the new equations in our paper is that if one knows the treatment effect conditioned on $Z$ (or an unbiased estimate, as we write), then we can  derive the treatment effect conditioned on the observation $X^*$ by averaging the treatment effect given $Z$ according to $P(Z|X^*)$. This is useful to justify why in equation (9) we form an estimate of $\tau$ for realizations of $Z$ according to $Z|X^*$, and then average these estimates over several realizations to estimate the average treatment effect. However, this does *not* tell us how to create an unbiased estimator of the treatment effect conditioned to $Z$; in particular, to compute the doubly robust estimator (9), this does not tell us how to correctly estimate the propensity score $\hat{e}$ and the response function $\hat{\mu}$ that are needed . For that purpose, we propose a heuristic method inspired by the approach of Kallus et al (2018) in the linear case, which may be biased (as discussed above).
> > >
> > > - Regarding experiments: as you suggested, we applied our methods on a close to real data set, namely the IHDP data set used as well by Shalit et al. (2017) and Louizos et al. (2017). We added a new Section 4.4 in the updated version of the paper which shows the good behavior of MissDeepCausal.
> > >
> > > Thank you again for your comments which help us improving the paper.

---

### Decision · Program_Chairs · 2019-12-19

**Decision:**

Reject

**Comment:**

This paper addresses the problem of causal inference from incomplete data. The main idea is to use a latent confounders through a VAE. A multiple imputation strategy is then used to account for missing values. Reviewers have mixed responses to this paper. Initially, the scores were 8,6,3. After discussion the reviewer who rated is 8 reduced their score to 6, but at the same time the score of 3 went up to 6. The reviewers agree that the problem tackled in the paper is difficult, and also acknowledge that the rebuttal of the paper was reasonable and honest. The authors added a simulation study which shows good results.

The main argument towards rejection is that the paper does not beat the state of the art. I do think that this is still ok if the paper brings useful insights for the community even though it does not beat the state fo the art. For now, with the current score, the paper does not make the cut. For this reason, I recommend to reject the paper, but I encourage the authors to resubmit this to another venue after improving the paper.